# Soil: the great connector of our lives now and beyond COVID-19

Rosa M. Poch[1], Lucia H. C. dos Anjos[2], Rafla Attia[3], Megan Balks[4], Adalberto Benavides-Mendoza[5], Martha M. Bolaños-Benavides[6], Costanza Calzolari[7], Lydia M. Chabala[8], Peter C. de Ruiter[9], Samuel Francke-Campaña[10], Fernando García Préchac[11], Ellen R. Graber[12], Siosiua Halavatau[13], Kutaiba M. Hassan[14], Edmond Hien[15], Ke Jin[16], Mohammad Khan[17], Maria Konyushkova[18], David A. Lobb[19], Matshwene E. Moshia[20], Jun Murase[21], Generose Nziguheba[22], Ashok K. Patra[23], Gary Pierzynski[24], Natalia Rodríguez Eugenio[25], Ronald Vargas Rojas[25]

[1]Universitat de Lleida, Spain
[2]Federal Rural University of Rio de Janeiro, Soils Dep. Brazil
[3]Ministère de l'Agriculture, des Ressources Hydrauliques et de la Pêche, Tunisia
[4]School of Science, University of Waikato, New Zealand
[5]Dept. of Horticulture, Universidad Autónoma Agraria Antonio Narro, Mexico
[6]Colombian Agricultural Research Corporation – AGROSAVIA, Colombia
[7]CNR - Institute of BioEconomy, Italy
[8]University of Zambia, Lusaka, Zambia
[9]Biometris, Wageningen University and Institute for Biodiversity and Ecosystem Dynamics, University of Amsterdam, The Netherlands
[10]Chilean Forestry Service, Secretary of Agriculture, Chile
[11]Facultad de Agronomía, Universidad de la República, Uruguay
[12]The Volcani Center, ARO, Rishon Letzion, Israel
[13]Ministry of  Agriculture, Tonga
[14]Ministry of Agriculture, Iraq
[15]Joseph Ki-Zerbo University, Ouagadougou, Burkina Faso
[16]Institute of Grassland Research, Chinese Academy of Agricultural Sciences, China
[17]Dept. of Soil and Environmental Sciences, The University of Agriculture, Peshawar, Pakistan
[18]Lomonosov Moscow State University, Eurasian Center for Food Security, Moscow, Russia
[19]University of Manitoba, Winnipeg, Canada
[20]University of Fort Hare, Faculty of Science and Agriculture, Alice, South Africa
[21]Graduate School of Bioagricultural Sciences, Nagoya University, Japan
[22]International Institute of Tropical Agriculture, Nairobi, Kenya
[23]ICAR-Indian Institute of Soil Science, Bhopal, India
[24]The Ohio State University, USA
[25]Global Soil Partnership, Food and Agriculture Organization of the United Nations

*Correspondence to*: Rosa M. Poch (rosa.poch@udl.cat)

**Abstract.** Humanity depends on the existence of healthy soils, both for the production of food and for ensuring a healthy, biodiverse environment, among other functions. COVID-19 is threatening food availability in many places of the world due

to the disruption of food chains, lack of workforce, closed borders and national lockdowns. As a consequence, more emphasis is being given to local food production, which may lead to more intensive cultivation of vulnerable areas and to soil degradation. In order to increase the resilience of populations facing this pandemic and future global crises, transitioning to a paradigm that relies more heavily on local food production on soils that are carefully tended and protected through sustainable management, is necessary. To reach this goal, the Intergovernmental Technical Panel on Soil (ITPS) of the Food and Agriculture Organization of the United Nations (FAO) recommends five active strategies: improved access to land, sound land use planning, sustainable soil management, enhanced research, and investments in education and extension.

> "The soil is the great connector of lives, the source and destination of all. It is the healer and restorer and resurrector, by which disease passes into health, age into youth, death into life. Without proper care for it we can have no community, because without proper care for it we can have no life."
>
> *Wendell Berry* (American novelist)

## 1 There is no food production without soil

The Coronavirus disease 2019 (COVID-19) pandemic is testing the ability of societies to survive an extreme global situation. Throughout history, humanity has gone through many cataclysms and disasters, but this is the first time in the Anthropocene that we face a crisis spanning the whole planet, drastically affecting everybody's lives. The global nature of the pandemic sheds a new light on how to ensure food security, which will increasingly depend on sufficient areas of fertile agricultural soils close to population centres. Healthy soils form our most necessary natural resource for food production, on which human existence is dependent (Vargas Rojas et al., 2016).

It is obvious that the COVID-19 pandemic will significantly reshape our lives well into the future, not just during this acute phase. In this vision paper, we, members of the Intergovernmental Technical Panel on Soils (ITPS[1]) of the FAO[2], share our understanding of the crucial role played by sustainable soil management in the new global reality. Indeed, appropriate soil management is imperative for solving and anticipating food security and nutrition requirements that governments and individuals will face in the post-pandemic world.

Our global society often relies on dense and interconnected networks of socio-economic relationships, which, in many respects, are far from efficient from an environmental point of view, and do not always support people's food and nutritional needs. Our health is directly linked to the quality of the environment in which we live and to the food we eat, as addressed by the 'One

---

[1] The ITPS is composed by 27 well-recognized soil scientists from all over the world (http://www.fao.org/global-soil-partnership/intergovernmental-technical-panel-soils/en/)

[2] Food and Agriculture Organization of the United Nations (http://www.fao.org/home/en/)

health'[3] initiative. The decline of environmental quality, caused by urban development and intensive agriculture have led in many places to significant losses of natural habitats and biodiversity (FAO et al., 2020). Human impacts on the environment exacerbate the appearance and spread of pathogens (FAO et al., 2020). Strong policies and actions that support healthy and productive soils are needed to assure global food security and sovereignty for urban and rural populations around the world
(Wittman and Blesh, 2017). It is vital that soils within or near cities are available, unpolluted, and managed sustainably, to ensure that they can contribute to safe local food production systems.

**2 Impact of COVID-19 pandemic on food and soil security**

It has been predicted that more people will die from hunger and starvation due to disrupted food production chains during the pandemic than from the disease itself (FSIN, 2020). Lack of workforce for harvesting and processing, restrictions in
transportation and movement of workers due to closed borders and national lockdowns, and shortages of production materials (e.g., seeds, fertilizers), have the potential, in some regions, to cause severe shortcomings in food availability. Even in areas where crop production is not disrupted, many cropping systems are monocultures designed for export, and cannot provide a well-rounded diet for local and national populations. Moreover, the disproportionate loss of older people due to the COVID-19 is a threat to secure food production because, in many parts of the world, the vast majority of farmers and people with
experience in agricultural management and understanding soil are over 50 years of age (e.g. IFAD, 2019; Eurostat, 2018). Thus, the pandemic may result in a considerable dearth of expertise in the future (Huynh et al., 2020), and thus reduced ability to continue to produce food and manage the soil sustainably everywhere around the globe.

The food supply in urban environments relies on longer and more complex food chains than in rural ones. The main effect of the pandemic in urban environments has been the excessive increase of food prices and food shortages, mainly in low-income
countries (Mukiibi, 2020; Competition Commission, 2020). In this situation, urban agriculture, which is already producing about 15-20% of the world's food supply (Lal, 2020b) is playing a critical role in cities with acute food shortages due to the COVID-19. Higher income urban environments are less affected and have undergone changes in food habits, as the increase of online food demand (Chang and Meyerhoefer, 2020). In rural areas of low-income countries, farmers are experiencing more severe problems, as difficulties to purchase seeds and fertilizers and to get produce to markets (World Farmers Organisation,
2020), besides financial barriers to access to credits. Some cases of illegal land clearing by companies while the locals were locked down have also been reported (Fox et al., 2020).

Additionally, the pressures of the COVID-19 crisis on food systems will also have a direct impact on soil security (Koch et al., 2013). International transport limitations will require a greater emphasis on local and national food production. In places where land suitable for agricultural use is limited, more intensive cultivation of already degraded soils and expansion of
agriculture to vulnerable areas could lead to increased soil degradation if not well managed (Willi et al., 2019). Degradation

---

[3] The One Health Initiative is a worldwide strategy for expanding interdisciplinary collaborations and communications in all aspects of health care for humans, animals and the environment (http://www.onehealthinitiative.com/)

results from depletion of soil carbon and nutrients, increased erosion, over-fertilization, soil salinization, soil pollution, and eventually, the loss of soils (Stocking, 2003), which are non-renewable resources. Soil degradation also results in increased atmospheric $CO_2$ emissions, which contribute to climate change (Olsson et al., 2019).

Perhaps the two main threats in the short term are soil pollution and nutrient depletion. Increase of urban agriculture is faced with the fact that contamination by heavy metals, organic pollutants, antibiotics, and petroleum products are among the major constraints limiting the use of urban soils for food production (Menefee and Hettiarachichi 2018). Moreover, there is evidence that the enormous quantity of disposable plastic gloves and face masks that get to the environment (soils and waters) could increase the accumulation of their related microplastics and fibers within a short time (Fadare and Okoffo, 2020; Aragaw, 2020). Another effect derives from the lack of access to fertilizers to small farmers in low-income countries, as it has already been reported in Thailand (Fox et al., 2020), which can easily lead to degradation by nutrient depletion.

### 3 Sustainable soil management to create and strengthen food systems

To prepare for a global disruption of food production from whatever cause, we suggest a general transition from the current emphasis on globalized food chains (King et al., 2017) to a more balanced approach that also includes well-rounded and diverse local, national, and regional food chains. Such a transition will help to build more resilient and secure societies in place, and is in the best interest of countries concerned about the welfare of their citizens. Countries and regions will need to identify ways to promote local food production, a circular agro-economy and recycling residues with potential for agricultural use (Jurgilevich et al., 2016).

Together with such a transition, soils must be carefully tended and protected (Fig. 1). It is essential that we invest in the sustainability of food production systems, and this implies caring for long term soil health so as to preserve soil structure, fertility, balanced organic matter and nutrients dynamics, biodiversity, and all the related soil ecosystem services (Lal, 2020a). Sustainable food production systems, in particular those that ensure food security for local populations, will require a transformation from land used for extensive large-scale monocultures to highly-diverse local agriculture, especially when they are near or within cities, in order to promote food resilient urban centres (Fig. 2). This change must go hand-in-hand with the strengthening of small farmers' capacities and soil awareness. It will also be increasingly important to ensure that populations do not exceed the carrying capacity of the lands on which they depend.

Several initiatives, since the onset of the pandemic and the beginning of food supply problems, have appeared around the world demonstrating that it is possible to improve food sovereignty thanks to the collaborative work of people. For instance, communities from Sabah (Malaysian Borneo), who are dependent on imports for 75% of their rice requirements are having problems sourcing rice from Vietnam and selling cash crops, have recovered their traditional practices and river culture to maintain their protein supply (Ong and Wilson, 2020). In Emilia Romagna (Italy), a region severely affected by the pandemic, farmer self-organisation has ensured the provision of local food making shorter chains (Diesner, 2020). Other initiatives, such as Slow Food Gardens in many countries in Africa, are strengthening rural small-scale producer communities and therefore ensuring food supply (Mukiibi, 2020). Rural communities in the developing world will require affordable and locally adapted

technologies to maintain soil health, while supporting diverse and well-rounded food production. For the long-term, this
requires governments and land owners to care about soils as a finite resource and to implement measures to prevent their
degradation (FAO, 2017; Batini et al., 2020).

Future agricultural policies should focus on coherent global agricultural regulations to avoid counterproductive market
interferences and to promote collaboration, and direct the transition to more diverse balanced systems. In this way, food
security can be achieved as much as possible on the basis of local food production chains. Governments should also support
research and educational areas that focus on food security (soil, water, seeding, management systems, processing, etc.) even
in the midst of global health challenges, since they will be even more important to building post COVID-19 food resilience.
Many countries have strategic grain reserves for meeting future national or international needs that can solve acute food
shortages. In much the same way, we need to create "Strategic Soil Reserves". By this, we do not mean locking away soils that
can then be released in the event of a future catastrophic loss of soil. We mean preserving, improving, rehabilitating and
140 protecting lands suitable for agriculture, especially our best peri-urban soils from urban development. Strategic Soil Reserves
can help solve long-term chronic food shortages.

Soil Security, as part of global, regional, national and local strategies, will ensure resilience in the face of such crises as we are
now experiencing. The overarching goal should be to achieve global food security and avoid enlarging gaps between societies.
The above is in line with the UN Sustainable Development Goals: No Poverty, Zero Hunger, Clean Water, Sustainable Cities,
Responsible Consumption and Production, Climate Action, and Life on Land. The Global Soil Partnership recommendations
as presented in the Voluntary Guidelines for Sustainable Soil Management (FAO, 2017) or the proposed RECSOIL (FAO,
2019a) mechanism to increase the resilience of soils by increasing soil organic carbon, in the frame of the Koronivia Joint
Work on Agriculture Roadmap (FAO, 2018), are now more relevant than ever.

The COVID-19 crisis has focused the worlds attention on the vulnerability of its food systems and the need to sustain food
production at a regional/country level. In view of the foregoing, the ITPS recommends five active strategies that will ensure
that each region/country has enough productive soil that can be managed sustainably to feed its population. These strategies
are access to land, sound land use planning, sustainable soil management, research, and education and extension.

• Access to land – It is necessary to revisit the national policies on land tenure to regulate international land ownership.
The access of local people to land, food and livelihood must be ensured by avoiding infringement of tenure rights by
155 business enterprises or states (FAO, 2012). In the same way, it is also critical to revisit the importance of small family
farms, which contribute greatly to maintenance of healthy soils and resilience of local communities in case of crisis.
• Sound land use planning - The need to preserve and improve local lands with agricultural potential and also to
convert or rehabilitate marginal areas when food production is needed, while protecting vital ecosystems, must
become a part of land use planning in all urban and rural development schemes. In particular, those soils that have a
160 high value for food production should be protected from land sealing due to urbanisation, infrastructure, or industrial
activities. This can be done, for example, by producing soil suitability maps for crops, using approaches such as the
Agro-Ecological Zone (AEZ) (FAO, 2002).

• Sustainable soil management - Site-specific conservation agriculture measures are needed to prevent land degradation and desertification. This will ensure the availability of productive soils for present and future generations. In particular, this requires taking appropriate actions to maintain, and where needed, improve soil fertility through integrated fertilization. Fertilization regimes should consider the nutritional requirements of the crops, the interactions of nutrients with the different soils and their intrinsic fertility, and the development of strategies that minimize soil pollution (FAO, 2019b). In particular, the development of food production in urban areas needs to be approached with caution and to include suitable testing, to ensure that existing soil pollution does not lead to toxic levels of contaminants in the produced food (Li et al., 2018).

• Research - Sustainable and resilient soil systems for food production will require increasing research efforts with new approaches and interdisciplinarity. The threats to sustainable soil management are not new, but research dealing with preserving soil quality for agriculture and reversal of soil degradation will be even more important. Much more research is needed on how to increase and maintain soil organic carbon (Torquebiau et al., 2018). Research in land sealing should be revisited to learn how to "de-seal" soils to bring them back into sustainable use for agriculture and forestry (Artmann, 2016). The focus on urban and peri-urban soils for food production must not be overlooked .

• Education and extension - The inclusion of soils in all levels of education curricula is necessary to increase awareness on the importance of soils in our lives. The strengthening of extension services, technology transfer and capacity-building programmes will support local farmers in applying sustainable practices. The development of mobile soil labs would help with fast diagnoses and with solving problems locally.

From its inception in 2012, the Global Soil Partnership of FAO is working in all these aspects through its five pillars of action (Soil management, Awareness raising, Promoting research, Information & data, Harmonization), which will acquire more importance in the light of the global crises.

Soils are a finite, non-renewable, multi-systemic source of life, and still they are easily overlooked in decision making acts and policies. A new post-pandemic reality should ensure that soil is recognised as the great connector and service provider that links our lives to all human needs of food, health and security (Moyer, 2020). Caring for soils is imperative to reduce the impacts of global disturbances, such as the current COVID-19 crisis.

**Author contributions**

R. M. Poch: Conceptualization, Writing - Original Draft Preparation, Review & Editing, Visualization L. H. C. dos Anjos: Writing - Original Draft Preparation, Review & Editing, R. Attia: Writing - Original Draft Preparation, Review & Editing, M. Balks: Writing - Original Draft Preparation, Review & Editing, A. Benavides-Mendoza: Writing - Original Draft Preparation, Review & Editing, M. M. Bolaños-Benavides: Writing - Original Draft Preparation, Review & Editing, C. Calzolari: Writing - Original Draft Preparation, Review & Editing, L. M. Chabala: Writing - Original Draft Preparation, Review & Editing, P. C. de Ruiter: Writing - Original Draft Preparation, Review & Editing, S. Francke-Campaña: Writing - Original Draft Preparation, Review & Editing, F. García Préchac: Writing - Original Draft Preparation, Review & Editing, E. R. Graber: Writing - Original

Draft Preparation, Review & Editing, S. Halavatau: Writing - Original Draft Preparation, Review & Editing, K. M. Hassan: Writing - Original Draft Preparation, Review & Editing, E. Hien: Writing - Original Draft Preparation, Review & Editing, K. Jin: Writing - Original Draft Preparation, Review & Editing, M. Khan: Writing - Original Draft Preparation, Review & Editing, M. Konyushkova: Writing - Original Draft Preparation, Review & Editing, D. A. Lobb: Writing - Original Draft Preparation, Review & Editing, M. E. Moshia: Writing - Original Draft Preparation, Review & Editing, J. Murase: Writing - Original Draft Preparation, Review & Editing, G. Nziguheba: Writing - Original Draft Preparation, Review & Editing, A. K. Patra: Writing - Original Draft Preparation, Review & Editing, G. Pierzynski: Writing - Original Draft Preparation, Review & Editing, N. Rodríguez Eugenio: Writing - Original Draft Preparation, Review & Editing, Visualization, R. Vargas Rojas: Writing - Original Draft Preparation, Review & Editing.

## Competing interests

The authors declare that they have no conflict of interest.

## Acknowledgments

This manuscript is an initiative of the ITPS. As such it has been elaborated, reviewed and endorsed by its members, in close collaboration with the GSP Secretariat.

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

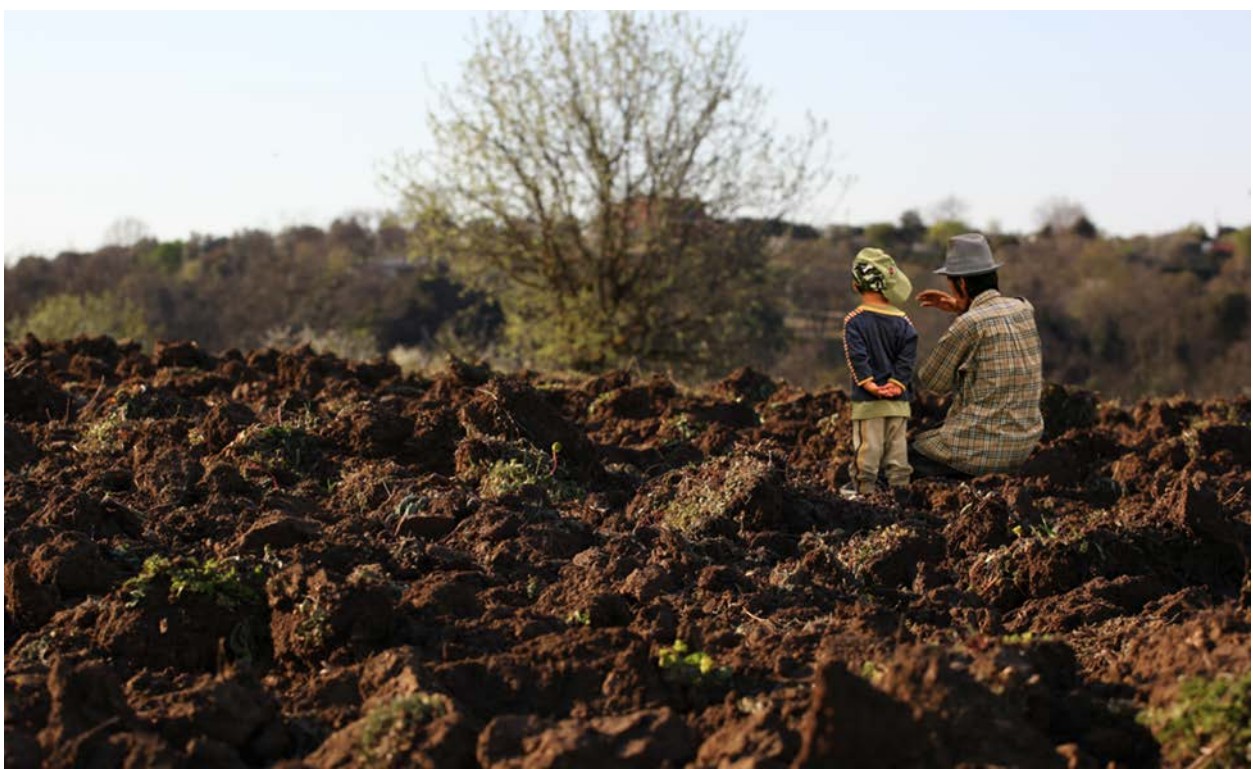

**Figure 1: In the heart of the sustainability concept, the connection with the land and respectness between generations lies. the basics and secrets about how to care about the soil are transferred from the older to the youth. ©Matteo Sala**

300

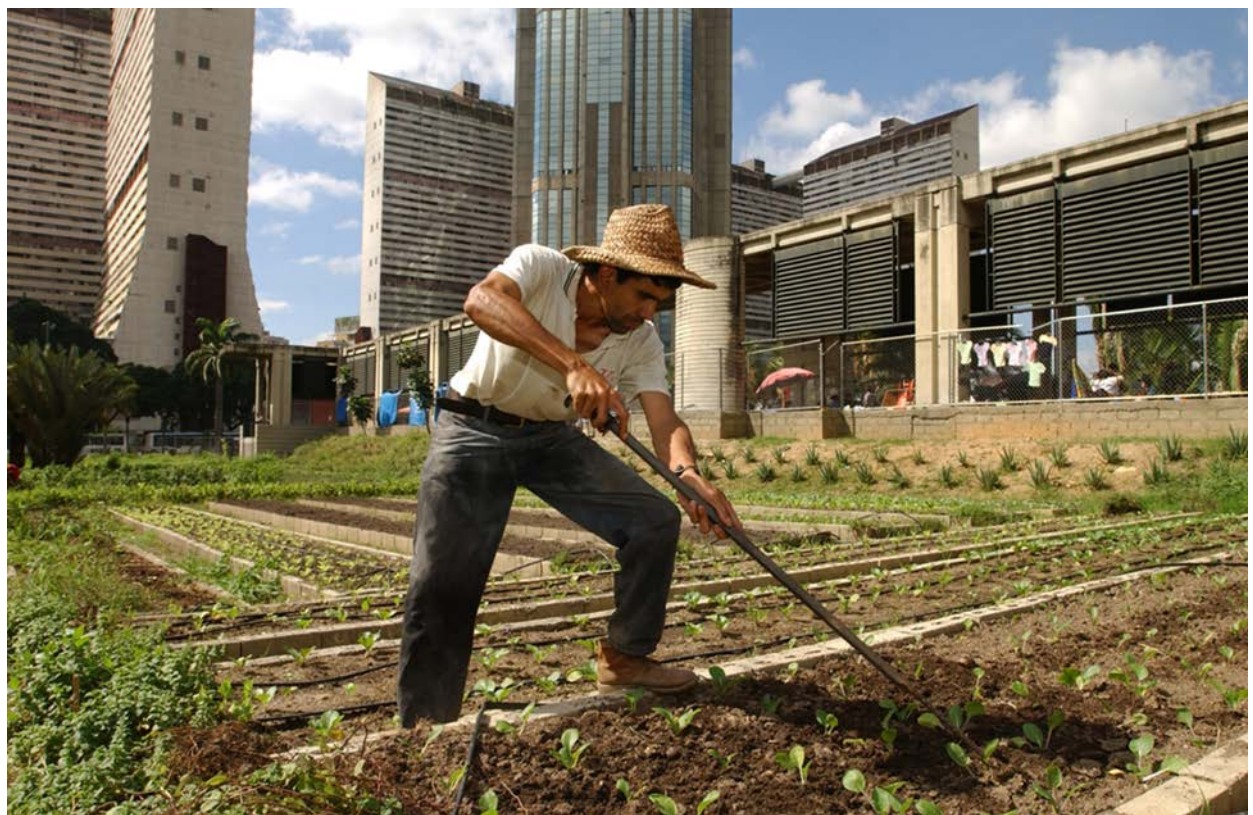

**Figure 2: Urban and peri-urban agriculture to improve nutrition and livelihoods of poor families as part of the Special Programme for Food Security (SPFS) in Caracas, Venezuela. ©FAO/Giuseppe Bizzarri**

305