# Peer review of "Soil: the great connector of our lives now and beyond COVID-19"

_SOIL, 2020_

## Referee Comment (RC1) · Anonymous Referee #1 · 27 Jul 2020

This text deals with the importance of soil for the future well-being of mankind. I do not see it as a scientific manuscript but more as a statement to change our prevailing economic system and swithc into something more sustainable. It is therefore quit difficult for me to mark it as either "go ahead with publishing" or "reject". In general I have doubts if this journal would be the right place to publish such a statement because I could not think of anyone who would disagree with the content which is obvious to the members of the soil science community in its generality. I would see the publication of such a manifesto in more publicly available media. I could imagine that publishing this manifesto after translation into national languages and moving it from a purely science oriented journal into different media with general public attention be more effective to convey its message. I therefore finally decided to suggest rejection not because I do

not agree with the content but because I would like to see these statements published at different national media. However, if the editors decide that such kind of messages would fit into the scope of 'Soil' I could quite happily live with it as well.

I rated the scientific significance poor because actually no new findings are presented.

---

## Referee Comment (RC2) · Anonymous Referee #2 · 28 Jul 2020

Reviewer comments on Poch et al "Soil: the great connector of our lives now and beyond COVID-19"

This article is written by members of the FAO Intergovernmental Technical Panel on Soil (ITPS). They draw attention to interruptions of food supply chains during the Covid-19 pandemic and state that this may promote greater reliance on local production of food. Many people would think this is a positive trend, especially for citizens in urban societies who have become increasingly separated from, and ignorant of, the sources of their food. However, these authors point to a possible downside of this trend, should it materialise. There is the risk that unsuitable soils will be used for intensive production and/or unsuitable management practices will be used, leading to degradation of soils and presumably negatively impact the long-term sustainability of food production. They

then list five strategies to combat this negative possibility, namely: improved access to land, sound land use planning, sustainable soil management, enhanced research, and investments in education and extension.

As a contribution to a scientific journal, this paper can easily be criticised for at least two reasons: 1. It contains no new scientific information. 2. The five strategies proposed, and the statements about the crucial role of soils in food production, have been well known for many decades. And even as a "forum" style or "opinion" article it has little merit because it contains nothing new.

However, I think there is an alternative view that carries some weight – but it will be an editorial decision as to their relative strengths. Although the statements in the paper are well known to soil scientists and agricultural practitioners, they may be less familiar to city-dwelling policy makers with little knowledge of soil, agriculture or agriculture/environment interactions. There can be value in having fundamental truths restated for new audiences. And it can be useful to have them contained in a citable scientific publication. Although it is unlikely that such policy makers will read the paper themselves, it is possible it will be seen and cited by their advisers.

If the paper is to be published in SOIL I suggest three additions, or rather expansions, that would make it more valuable. It is currently written in the most general terms, with no specifics. Unless the authors can add some "flesh to the bones" I do not think it is particularly useful as a new publication. Of course, no one can disagree with its main thrusts regarding the value of soils and the importance of appropriate land use planning and management practices. But these points are set out in numerous reports from FAO, national and regional governments, NGOs, guidance materials for farmers and in textbooks.

1. It is unclear whether the authors are referring primarily to the situation in higher-income countries and/or large-scale farming or to small-scale farming in low-income countries; or perhaps to both? It would be helpful to be more specific and include

comments on the issues relevant in these differing situations. Almost certainly the social and economic factors of importance will be different, though perhaps there are some generic points to be made. 2. The authors mention the common causes of soil degradation including "depletion of soil carbon and nutrients, increased erosion, over-fertilization, soil salinization, soil pollution..." It would be helpful to comment on which of these is most likely in different situations. Even give a few examples; of course, these will not be exhaustive but would give some substance to the article, as opposed to rather vague generalities. Examples from specific regions or situations could be useful as early warning of potential dangers elsewhere. 3. The strategies to combat soil degradation (and wider impacts on ecosystems) are all well known, have been stated and discussed for decades but sadly, in many situations, have been either ignored or even reversed. It would be helpful if the authors mentioned some key barriers to their implementation and give some fresh ideas on ways of overcoming them. Are there any good examples that could be pointers for addressing these issues elsewhere?

---

## Referee Comment (RC3) · Anonymous Referee #3 · 29 Jul 2020

A very important part of this document is the statement that "It is vital that soils in or near cities are available, uncontaminated and sustainably managed, to ensure that they can contribute to the safety of local food production systems". Thoughts that are also part of the essential approaches of the "Slow down" movement that is practiced today by various communities in many countries and perhaps examples of positive achievements of this could be referred and/or analyzed.

However, to give more weight to the article and make it more beneficial and interesting, precisely because the "soils are finite and non-renewable", I propose that the authors go one step further in the 5 strategies proposed to delve into how the ITPS considers that firmly and consistently progress can be made towards achieving these five strategies: "Access to land, rational land use planning, sustainable land management,

research, education, and extension."

How do you propose to move forward effectively and prioritized to achieve it?

I think that the main contribution of this article can be to create opinions and generate the necessary discussion on important issues for soil, agriculture, and its impact on humanity, such as what production and conservation alternatives are proposed by a world-class organization such as FAO, indicating how to implement its five strategies.

I share the author's vision when his text indicates that: "Indeed, appropriate soil management is imperative for solving and anticipating food security and nutrition requirements that governments and individuals will face in the post-pandemic world." However, I want to make a comment strongly provocative but equally essential for those of us in science, whose must begin to offer clarity about the limit of growth in the development (Meadows, 1977) that the thermodynamic state of the planet's ecosystem balances imposes on us (Progogine, 1974) as an indispensable part of the "new normality", and ask:

Is it not time to recognize that the human population cannot, and should not continue to grow at the current rate, and initiate powerful awareness-raising actions in this regard? Should not we start talking and writing about the need for conscious population control to really guarantee soil, food, water, and life sustainability?

---

## Author Comment (AC1) · 4 Aug 2020

We thank Referee#1 for his/her comments and considerations. You are right: our manuscript does not contain new scientific information, since it was intended as a Forum article. According to the SOIL policy, it "should stimulate an open debate by presenting new ideas and views of soil as part of the larger Earth system. As such, they must strive to be a point of departure for future work. Purely speculative contributions are discouraged." The five strategies we propose are not new. But we disagree about the little merit since we give reasons for them being useful (even crucial, to our understanding) to increase the resilience of societies in front of pandemics. After reviewing several publications on the effects of COVID in different sectors (as agriculture, hydrology, economics, world trade, sociology...) the same arguments could be stated,

saying that there is nothing "scientifically" new (unless we are dealing with the virus and disease itself, which are indeed new), but analysed from a different viewpoint. We believe that the statements at the end of the manuscript can be used as guidelines for policy proposals to mitigate the effects of pandemics and global crises, stressing the central role soils should have. We are convinced that these "well-known" truths have to be restated under the scope of COVID-19, in the same way e.g. the IPCC is repeating publication after publication that GHG emissions have to be reduced, and they are not blamed for not being new. We also think that the possibility to publish this manuscript as Forum paper in an open access, comprehensive and transversal journal as SOIL will be an added value for the possibility to become a citable paper and also for the inputs of reviews and comments that will surely come from it. Another high point is the opportunity to bring new interdisciplinary research integrating urban development and agriculture sustainability. It is a common fact that many of the media that deal with translating scientific articles to the non-scientific public prefer to have as sources subjects that are published first in a journal. This is understandable, since it will carry the benefit of the peer review system, and thus insure the information as from a cited publication. In this aspect it is of relevance that journals also divulge such a message, especially considering this pandemic condition, which will lead to major changes on production of knowledge and investments in scientific research. Soil plays a major part in the process of changing agriculture models and although there are not new findings in this paper, it is a new approach and that is relevant to achieve a future where food production is sustainable. As mentioned by the reviewer, once it is published, it may be translated in many languages, with the collaboration of ITPS members, and in this way further divulge the ideas posted in this paper and the Journal as well. Thus, we believe that in some critical moments a science-oriented journal would also benefit from documents such as this one.

---

## Author Comment (AC2) · 4 Aug 2020

We thank Referee#2 for his/her comments and considerations. You are right: our manuscript does not contain new scientific information, since it was intended as a Forum article. According to the SOIL policy, it "should stimulate an open debate by presenting new ideas and views of soil as part of the larger Earth system. As such, they must strive to be a point of departure for future work. Purely speculative contributions are discouraged."

It is true that the five strategies are not new. But we disagree about the little merit as a forum article, since we give reasons for them being useful (even crucial, to our understanding) to increase the resilience of societies in front of pandemics. After re-

viewing several publications on the effects of COVID in different sectors (as agriculture, hydrology, economics, world trade, sociology…) the same arguments could be stated, saying that there is nothing "scientifically" new (unless we are dealing with the virus and disease itself, which are indeed new), but analysed from a different viewpoint. The statements at the end of the manuscript can be used as guidelines for policy proposals to mitigate the effects of pandemics, stressing the central role soils should have. We believe that these "well-known" truths have to be restated under the scope of COVID-19, in the same way e.g. the IPCC is repeating publication after publication that GHG emissions have to be reduced, and they are not blamed for not being new.

We also think that the possibility to publish this manuscript as forum paper in an open access, comprehensive and transversal journal as SOIL will be an added value, as you suggest, for the possibility to become a citable paper and also for the inputs of reviews and comments that will surely come from it. Another high point is the opportunity to bring new interdisciplinary research integrating urban development and agriculture sustainability.

The referee points out that "It is unclear whether the authors are referring primarily to the situation in higher income countries and/or large-scale farming or to small-scale farming in low-income countries; or perhaps to both? It would be helpful to be more specific and include comments on the issues relevant in these differing situations. Almost certainly the social and economic factors of importance will be different, though perhaps there are some generic points to be made."

A/ Thank you for this suggestion. Our comments are referring to generic situations, but indeed there are differences not only regarding high/low income countries, but also re-garding urban/rural environments and different land tenure systems. We have enlarged this part giving examples of these contrasting environments:

"The food supply in urban environments relies on longer and more complex food chains than in rural ones. The main effect of the pandemic in urban environments has been the

excessive increase of food prices and food shortages, mainly in low-income countries (Mukiibi, 2020; Competition Commission, 2020). In this situation, urban agriculture, which is already producing about 15-20% of the world's food supply (Lal, 2020b) is playing a critical role in cities with acute food shortages due to the COVID-19. Higher income urban environments are less affected and have undergone changes in food habits, as the increase of online food demand (Chang and Meyerhoefer, 2020). In rural areas of low-income countries, farmers are experiencing more severe problems, as difficulties to purchase seeds and fertilizers and to get produce to markets (World Farmers Organisation, 2020), besides financial barriers to access to credits. Some cases of illegal land clearing by companies while the locals were locked down have also been reported (Fox et al., 2020)."

A second suggestion of Referee#2 is that "The authors mention the common causes of soil degradation including "depletion of soil carbon and nutrients, increased erosion, overfertilization, soil salinization, soil pollution..." It would be helpful to comment on which of these is most likely in different situations. Even give a few examples; of course, these will not be exhaustive but would give some substance to the article, as opposed to rather vague generalities. Examples from specific regions or situations could be useful as early warning of potential dangers elsewhere."

A/ It is evident that COVID-19 is not generating all types of soil degradation in the short term, and it is uncertain -as we state in the manuscript- what will be the mid-long-term effects of the change in land use caused by the need of food production near cities. As you are suggesting, we have made explicit two consequences/problems of the pandemic on soils: pollution and lack of fertilisation:

"Perhaps the two main threats in the short term are soil pollution and nutrient depletion. Increase of urban agriculture is faced with the fact that contamination by heavy metals, organic pollutants, antibiotics, and petroleum products are among the major constraints limiting the use of urban soils for food production (Menefee and Hettiarachichi 2018). Moreover, there is evidence that the enormous quantity of disposable plastic

gloves and face masks that get to the environment (soils and waters) could increase the accumulation of their related microplastics and fibers within a short time (Fadare and Okoffo, 2020; Aragaw, 2020). Another effect derives from the lack of access to fertilizers to small farmers in low-income countries, as it has already been reported in Thailand (Fox et al., 2020), which can easily lead to degradation by nutrient depletion."

The last suggestion of Referee#2 is the following: "The strategies to combat soil degradation (and wider impacts on ecosystems) are all well known, have been stated and discussed for decades but sadly, in many situations, have been either ignored or even reversed. It would be helpful if the authors mentioned some key barriers to their implementation and give some fresh ideas on ways of overcoming them. Are there any good examples that could be pointers for addressing these issues elsewhere?"

A/ It is true that the strategies that have been repeatedly proposed have many barriers that prevent their implementation, but it is out of this Forum article to focus on them since they are also well known. Regarding the solutions, even after the first submission of this manuscript, several initiatives have been reported around the world to improve the resilience of food supply regarding soils –the starting point. We have added a paragraph illustrating them:

"Several initiatives, since the onset of the pandemic and the beginning of food supply problems, have appeared around the world demonstrating that it is possible to improve food sovereignty thanks to the collaborative work of people. For instance, communities from Sabah (Malaysian Borneo), who are dependent on imports for 75% of their rice requirements are having problems sourcing rice from Vietnam and selling cash crops, have recovered their traditional practices and river culture to maintain their protein supply (Ong and Wilson, 2020). In Emilia Romagna (Italy), a region severely affected by the pandemic, farmer self-organisation has ensured the provision of local food making shorter chains (Diesner, 2020). Other initiatives, such as Slow Food Gardens in many countries in Africa, are strengthening rural small-scale producer communities and therefore ensuring food supply (Mukiibi, 2020)."

---

## Author Comment (AC3) · 4 Aug 2020

We thank Referee#3 for his/her comments. His/her first question is: "How do you propose to move forward effectively and prioritized to achieve it? I think that the main contribution of this article can be to create opinions and generate the necessary discussion on important issues for soil, agriculture, and its impact on humanity, such as what production and conservation alternatives are proposed by a world-class organization such as FAO, indicating how to implement its five strategies."

A/ As you indicate, those initiatives in the frame of "slow food" movements around the world are being proved successful to ensure resilience of the societies around the world in the present crisis, and therefore we have added some examples of them:

[Figure]

"Several initiatives, since the onset of the pandemic and the beginning of food supply problems, have appeared around the world demonstrating that it is possible to improve food sovereignty thanks to the collaborative work of people. For instance, communities from Sabah (Malaysian Borneo), who are dependent on imports for 75% of their rice requirements are having problems sourcing rice from Vietnam and selling cash crops, have recovered their traditional practices and river culture to maintain their protein supply (Ong and Wilson, 2020). In Emilia Romagna (Italy), a region severely affected by the pandemic, farmer self-organisation has ensured the provision of local food making shorter chains (Diesner, 2020). Other initiatives, such as Slow Food Gardens in many countries in Africa, are strengthening rural small-scale producer communities and therefore ensuring food supply (Mukiibi, 2020)."

In relation to your suggestion to move forward from the 5 proposed strategies, the Global Soil Partnership of the FAO is structured in 5 pillars of action from its beginning in 2012: (1) Promote sustainable management of soil resources for soil protection, conservation and sustainable productivity; (2) Encourage investment, technical cooperation, policy, education awareness and extension in soil; (3) Promote targeted soil research and development focusing on identified gaps and priorities and synergies with related productive, environmental and social development actions; (4) Enhance the quantity and quality of soil data and information: data collection (generation), analysis, validation, reporting, monitoring and integration with other disciplines; (5) Harmonization of methods, measurements and indicators for the sustainable management and protection of soil resources. This means that FAO is already involved and working in most of the strategies. We included a sentence referring to it:

"From its inception in 2012, the Global Soil Partnership of FAO is working in all these aspects through its five pillars of action (Soil management, Awareness raising, Promoting research, Information & data, Harmonization), which will acquire more importance in the light of the global crises."

The second question of Referee#3 is: "Is it not time to recognize that the human population cannot, and should not continue to grow at the current rate, and initiate powerful awareness-raising actions in this regard? Should not we start talking and writing about the need for conscious population control to really guarantee soil, food, water, and life sustainability?"

A/ Your question is indeed controversial, and certainly we agree that human population growth is the main aspect of agricultural sustainability and food security. But it has ecological and ethical implications whose discussion falls out of the initial scope of our manuscript; moreover, this is a sensitive subject when so many lost family members and friends all over the world, due to the pandemic. Nevertheless, we have added a sentence pointing out the importance of this issue in section 3: "It will also be increasingly important to ensure that populations do not exceed the carrying capacity of the lands on which they depend."

Some researchers estimated 20 y ago that world soils can still feed an increasing world's population if well managed (see, e.g. Eswaran, H., F. Beinroth, and P. Reich. 1999. Global land resources and population supporting capacity. Am. J. Alternative Agric. 14:129-136) but boundary conditions are very variable and uncertain. But, we believe that the ITPS five active strategies, to have enough productive soil that can be managed sustainably to feed the country's population, access to land, sound land use planning, sustainable soil management, research, and education and extension, also carry the benefit of human population control, though education and awareness of the limits of the land to produce in a sustainable way. Many experiences from FAO and other organizations show that one of the most effective means of raising the general level of development and promoting sustainable development is the education and empowering the woman in poor developed countries, with significant reduction of fertility rates (https://www.worldbank.org/en/topic/girlseducation) (http://www.fao.org/3/W6038E/w6038e02.htm)

---

## Referee Comment (RC4) · Anonymous Referee #3 · 7 Aug 2020

Thank the authors for your answer.

I believe that mentioning some of these cases on local and successful production strategies would enrich your document by making the reader aware of real alternatives that allow us to guarantee the resilience of societies around the world in the current crisis.

Of course, this will be at the discretion of the editor. Although your information responds to my comment, this is not an absolute requirement on my part but an attempt to bring the reader closer to those tangible examples of successful experiences.

As an extra comment, Slow Down is a broad movement of which Slow Food is only a part, and its importance lies in the fact that it seeks to promote all kinds of sustainable,

and empowering actions of the communities and that offers successful examples of organization in different countries, continents, and areas of social interaction.

Thank you for your proposal to insert the following two comments:

1) "From its inception in 2012, the Global Soil Partnership of FAO is working in all these aspects through its five pillars of action (Soil management, Awareness raising, Promoting research, Information & data, Harmonization), which will acquire more importance in the light of the global crises."

A) I consider it necessary because it makes clear that beyond the proposals there is a Plan and actions in this regard to carrying them out.

2) "It will also be increasingly important to ensure that populations do not exceed the carrying capacity of the lands on which they depend."

A) Indeed, talking about human population growth IS, and it has always been a sensitive and controversial topic for many different reasons. However, and precisely before all the pain and damage caused by the current pandemic, we must consider the changes that humanity requires today to recognize everything that needs to be modified to move towards true sustainability and in the prevention of future catastrophes.

Everything can be said if you look for a way to do it to be heard, and the inclusion of the phrase "It will also be increasingly important to ensure that populations do not exceed the carrying capacity of the lands on which they depend.", is sufficiently clear to those who they know how to read.

In this way, we do not ignore our knowledge about that the control of human population growth is the principal aspect of agricultural sustainability and food security, and, if on the contrary, we are calling to the reflection about it.

Finally, I agree, education is the most effective means of raising the general level of development and promoting sustainability, but also are educating when we write with that intention towards the reader. Hence the importance of writing about what despite,

being controversial, needs to be said.

However, as I wrote before, it will be the editor's judgment that ultimately determines what is appropriate.

---

## Author Comment (AC4) · 26 Aug 2020

Thank you for your useful and constructive comments. Indeed we are dealing with not easy issues that go beyond strict soil science and demonstrate that soils are affecting social behaviour and are closely linked to the evolution of humanity. We are facing now one of these crosspoints where the societies must make the good choices regarding food production and sustainability in the mid and long term. Let's hope for the best.

---

## Author Response (AR1)

**Answers to Comments on soil-2020-38-comments-toauthor**

*Page: 2*
*Author: jquin Subject: Sticky Note Date: 27/8/2020 10:08:26*
*what about Climate change? early 20th century Flu pandemics?*
**A/** Yes, you are right, there have been other pandemics and climate change is ongoing, but Covid-19 is the first one immediately affecting all aspects of social functioning, trade, daily living among other sectors, everywhere. The Flu pandemic occurred at a time without globalization – at least not at the level we have now; and Climate change is occurring gradually and not affecting the whole planet in the same way nor intensity – although its consequences will be indeed more severe in a long term. We suggest to change the sentence to:
"… but this is the first time in the Anthropocene that we face a crisis spanning the whole planet, drastically affecting everybody's lives."

*suggest reword to 'the global nature of the pandemic...'*
**A/** Sentence reworded.

*Page: 3*
*Author: jquin Subject: Sticky Note Date: 27/8/2020 10:12:25*
*Are these statements all part of the one health initiative? If not, they are disconnected and the narrative needs flow between them to put forward an argument. If they are consider reworking to make this clearer.*
**A/** They are not part of the one health initiative, but they are related to soil health. We have reworked these sentences to link the arguments as you suggest:
"Our health is directly linked to the quality of the environment in which we live and to the food we eat, as addressed by the 'One health' initiative. The decline of environmental quality, caused by urban development and intensive agriculture have led in many places to significant losses of natural habitats and biodiversity (FAO et al., 2020). Human impacts on the environment exacerbate the appearance and spread of pathogens (FAO et al., 2020)."

*Author: jquin Subject: Sticky Note Date: 27/8/2020 10:14:12*
*Not sure that we have seen this - any evidence? if not tone down to 'have the potential to..'*
**A/** There are local evidences for that, but they are actually not generalized. We changed the sentence to:
"… shortages of production materials (e.g., seeds, fertilizers), have the potential, in some regions, to cause severe shortcomings in food availability."

*Author: jquin Subject: Inserted Text Date: 27/8/2020 10:14:26*
*in the future*
**A/** Done.

*Author: jquin Subject: Sticky Note Date: 27/8/2020 10:15:32*
*reference required. Is there conclusive proof for this?*
**A/** Reference added:

Olsson, L., Barbosa, H., Bhadwal, S., Cowie, A., Delusca, K., Flores-Renteria, D., Hermans, K., Jobbagy, E., Kurz, W., Li, D., Sonwa, D.J. and Stringer, L.: 2019: Land Degradation. In: P.R. Shukla, J. Skea, E. Calvo Buendia, V. Masson-Delmotte, H.-O. Pörtner, D. C. Roberts, P. Zhai, R. Slade, S. Connors, R. van Diemen, M. Ferrat, E. Haughey, S. Luz, S. Neogi, M. Pathak, J. Petzold, J. Portugal Pereira, P. Vyas, E. Huntley, K. Kissick, M. Belkacemi and J. Malley, (Eds.), Climate Change and Land: an IPCC special report on climate change, desertification, land degradation, sustainable land management, food security, and greenhouse gas fluxes in terrestrial ecosystems.
https://www.ipcc.ch/srccl/chapter/chapter-4/, 2019.

*Page: 4*
*Author: jquin Subject: Sticky Note Date: 27/8/2020 10:18:27*
*maybe also protecting our best peri urban soils from urban development?*
**A/** Yes, it is implicitely stated, but we have rephrased it:
"We mean preserving, improving, rehabilitating and protecting lands suitable for agriculture, especially our best peri-urban soils from urban development."

*Author: jquin Subject: Sticky Note Date: 27/8/2020 10:23:11*
*I wonder if you could have a stronger introduction to these points stressing the link with covid?*
*the Covid-19 crisis has focused the worlds attention on the vulnerability of its food systems and the need to sustain food production at a regional/country level...*
**A/** Thank you for your suggestion that improves the flow of the text.

---

## Editor Decision (ED1)

[revised manuscript text omitted]